# Estimating the impact of school closures on the COVID-19 dynamics in 74 countries: A modelling analysis

Romain Ragonnet[1]*, Angus E. Hughes[1], David S. Shipman[1], Michael T. Meehan[2], Alec S. Henderson[2], Guillaume Briffoteaux[3], Nouredine Melab[4], Daniel Tuyttens[3], Emma S. McBryde[5], James M. Trauer[1]

1 School of Public Health and Preventive Medicine, Monash University, Melbourne, Australia, 2 Australian Institute of Tropical Health and Medicine, James Cook University, Douglas, Australia, 3 Mathematics and Operational Research Department, University of Mons, Mons, Belgium, 4 CNRS CRIStAL, Inria Lille, Université de Lille, Lille, France, 5 University of Queensland Centre for Clinical Research (UQCCR), University of Queensland, Brisbane, Australia

* romain.ragonnet@monash.edu

**Data Availability Statement:** All data and codes used and generated by this analysis are publicly

## Abstract

### Background

School closures have been a prominent component of the global Coronavirus Disease 2019 (COVID-19) response. However, their effect on viral transmission, COVID-19 mortality and health care system pressure remains incompletely understood, as traditional observational studies fall short in assessing such population-level impacts.

### Methods and findings

We used a mathematical model to simulate the COVID-19 epidemics of 74 countries, incorporating observed data from 2020 to 2022 and historical school closure timelines. We then simulated a counterfactual scenario, assuming that schools remained open throughout the study period. We compared the simulated epidemics in terms of Severe Acute Respiratory Syndrome Coronavirus 2 (SARS-CoV-2) infections, deaths, and hospital occupancy pressure. We estimated that school closures achieved moderate to significant burden reductions in most settings over the period 2020 to 2022. They reduced peak hospital occupancy pressure in nearly all countries, with 72 out of 74 countries (97%) showing a positive median estimated effect, and median estimated effect ranging from reducing peak hospital occupancy pressure by 89% in Brazil to increasing it by 19% in Indonesia. The median estimated effect of school closures on COVID-19 deaths ranged from a 73% reduction in Thailand to a 7% increase in the United Kingdom. We estimated that school closures may have increased overall COVID-19 mortality (based on median estimates) in 9 countries (12%), including several European nations and Indonesia. This is attributed to changes in population-level immunity dynamics, leading to a concentration of the epidemic during the Delta variant period, alongside an upward shift in the age distribution of infections. While our estimates were associated with significant uncertainty, our sensitivity analyses exploring the impact of social mixing assumptions revealed robustness in our country-specific conclusions. The

available in a GitHub repository (github.com/monash-emu/covid19_school_closures).

**Funding:** RR was supported by an Investigator Grant from the Australian National Health and Medical Research Council (GNT2025844, https://www.nhmrc.gov.au/). JMT was supported by a Discovery Early Career Researcher Award from the Australian Research Council (DE230100730, https://www.arc.gov.au/). The Epidemiological Modelling Unit at the School of Public Health and Preventive Medicine (EMU) was supported by a Rapid Research Digital Infrastructure COVID-19 grant from the Medical Research Future Fund during 2021 and 2022 (RRDHI000027, https://www.health.gov.au/our-work/medical-research-future-fund). The funders had no role in the data collection, study design, conduct, reporting or decision to publish.

**Competing interests:** The authors have declared that no competing interests exist.

main study limitations include the fact that analyses were conducted at the national level, whereas school closure policies often varied by region. Furthermore, some regions, including Africa, were underrepresented due to insufficient data informing the model.

## Conclusions

Our analysis revealed nuanced effects of school closures on COVID-19 dynamics, with reductions in COVID-19 impacts in most countries but negative epidemiological effects in a few others. We identified critical mechanisms for consideration in future policy decisions, highlighting the unpredictable nature of emerging variants and potential shifts in infection demographics associated with school closures.

---

## Author summary

### Why was this study done?

- Most countries closed schools during the early stages of the Coronavirus Disease 2019 (COVID-19) pandemic in an attempt to slow the virus spread.

- The extent, duration, and timing of school closures varied greatly between countries.

- There was a need to analyse systematically how these closures affected virus transmission, healthcare systems, and overall pandemic outcomes across different settings.

### What did the researchers do and find?

- We used a mathematical model to simulate the transmission of COVID-19 from 2020 to 2022, incorporating the effects of school closures in 74 countries.

- We also simulated a counterfactual scenario to estimate what might have happened if schools had stayed open, focusing on infections, hospitalisations, and death.

- We estimated that school closures significantly reduced the COVID-19 burden in most countries but might have increased cumulative mortality in a few cases.

### What do these findings mean?

- While school closures were effective in many cases, their overall impact varied widely by setting.

- Our findings underscore the importance of a tailored approach to such interventions, considering both their immediate and longer-term impacts on local epidemics.

- Future multidisciplinary studies could expand upon our findings by evaluating the broader impacts of school closures, including not only COVID-19 but also other health issues, economic consequences, and educational outcomes.

- The main limitations of the study are that the analyses were done at the national level, even though school closure policies varied between regions. Additionally, we were only able to cover a small number of countries in certain regions, including Africa, because there was insufficient data to inform our analysis.

## Introduction

The global outbreak of Coronavirus Disease 2019 (COVID-19), caused by the Severe Acute Respiratory Syndrome Coronavirus 2 (SARS-CoV-2), prompted governments to implement a range of public health and social measures in an attempt to mitigate its impact. Among these interventions, school closures emerged as a widely implemented measure in an attempt to prevent transmission and reduce pressure on health systems. Previously employed in the context of influenza pandemic control, this intervention had been recommended as a precautionary measure in the early stages of a pandemic, despite the lack of sufficient evidence to endorse any specific school closure policy [1]. School closures for COVID-19 were particularly prevalent during the first 18 months of the pandemic, as populations could not be protected by widespread vaccination.

This intervention has been a topic of debate and investigation since the early days of the COVID-19 pandemic [2,3]. While the potential benefits of keeping schools open for educational and socioemotional reasons are evident, concerns about schools acting as amplifiers of viral transmission drove policymakers to adopt temporary closures as a preventive measure. The multifaceted social consequences of school closures, ranging from disruptions to learning routines and the exacerbation of educational inequalities to potential economic impacts [4], necessitate a thorough appraisal of their effects on the dynamics of COVID-19 epidemics.

The epidemiological impact of school closures at the population-level cannot be directly measured using traditional approaches such as observational studies. This is because measuring epidemic indicators under a counterfactual scenario, where schools would have remained open, is not feasible. Furthermore, the impact of school closures is complicated by the presence of confounding factors and collinearity with other non-pharmacological interventions that were introduced concurrently [5]. As a result, the effectiveness of school closures remains uncertain. Mathematical modelling offers a viable method for conducting such estimations by carefully capturing the characteristics of the epidemics and the different interventions in place, as well as simulating counterfactual scenarios.

Previous studies investigating the effect of school closures on COVID-19 have primarily focused on the immediate impact of the intervention on transmission dynamics, often through estimates of the reproductive number. While this approach provides valuable insights into the short-term effects, it overlooks the broader, long-term consequences of such policies. To evaluate school closure interventions comprehensively, it is crucial to consider their impact over extended periods, accounting for changes in population immunity and their effects on multiple health indicators beyond just infection rates, such as deaths and hospitalisations.

Additionally, much of the existing research has concentrated on single-country or single-region analyses. This limited scope does not account for the expected contextual variations in the effectiveness of school closures, which can be influenced by differences in demographic and behavioural factors across settings.

Our study aimed to address these gaps by providing a more holistic assessment of school closures' long-term effects across diverse settings. We investigated the complex relationship

between school closures and the spread of COVID-19 in 74 countries over the three-year period 2020 to 2022, providing a comprehensive assessment of the impact of this intervention on SARS-CoV-2 transmission, as well as COVID-19–induced mortality and hospital occupancy pressure. By analysing a wide range of epidemiological and demographic settings, we aimed to gain deeper insights into the effectiveness of this intervention in order to inform future pandemic control around the world.

## Methods

### Country inclusion and data sources

Included countries were required to have publicly available data on: population size by age (from the United Nations' Population Division) [6]; population mobility over time (Google) [7]; emergence times for both Delta and Omicron variants (GISAID database) [8]; periods of school closures in primary and secondary education (UNESCO) [9]; COVID-19 deaths over time (World Health Organization) [10]; and COVID-19 vaccine coverage over time (Our World in Data database) [11]. Additionally, we excluded countries that had reported no more than 5,000 cumulative COVID-19 deaths by 31 December 2022, as it is difficult to capture such small epidemics realistically. One further country was excluded because no school closure periods were reported during the analysis period (Belarus) [9]. Although we used seroprevalence data extracted from the SeroTracker database to inform the models when accessible, the availability of these data was not an inclusion criterion [12]. We ultimately included 74 countries in the analysis, comprising 30 from Europe, 21 from Asia, 10 from South America, 7 from North America, 5 from Africa, and 1 being Australia (Fig 1).

To ensure the robustness and reliability of the analyses, we applied strict inclusion criteria and used only available data points for model calibration, such that imputation for missing data was not required. Additionally, we visually inspected the data for each country to identify any unrealistic features, such as the emergence of variants of concern prior to their global recognition or a decrease in vaccination coverage over time.

### Model description

We designed a mathematical model of SARS-CoV-2 transmission, building upon the core methodologies employed in our prior studies that utilised mechanistic models [13–15], and incorporating a nonmechanistic component inspired by another COVID-19 modelling study [16]. Our model was directly informed by the country-specific data mentioned in the previous section.

Our compartmental model, governed by ordinary differential equations, captured a sequence of infection-related states: Susceptible (not previously infected), Exposed (recently infected with SARS-CoV-2 but not yet in the active phase of disease), Actively Diseased, and Recovered. We used 4 Exposed compartments and 4 Active Disease compartments arranged in series to achieve realistic time distributions for the periods spent in each of these states, with alternative assumptions regarding the number of serial compartments explored in sensitivity analyses (Section B.1.6 in S1 Appendix). All Active Disease compartments were assumed to be infectious, while the last 2 Exposed compartments were considered only partially infectious. All model compartments were stratified by age (0 to 14 years, 15 to 24 years, 25 to 49 years, 50 to 69 years, 70 years and over) and vaccination status (at least 2 doses received or not) using time-varying vaccination rates informed by country-specific coverage over time. We further stratified all compartments except for the Susceptible compartment by viral strain in order to capture 3 viral variants explicitly: Wild-type virus, Delta variant, and Omicron variant. We included all the variants preceding the emergence of Delta (e.g., Alpha, Beta) under the "Wild-

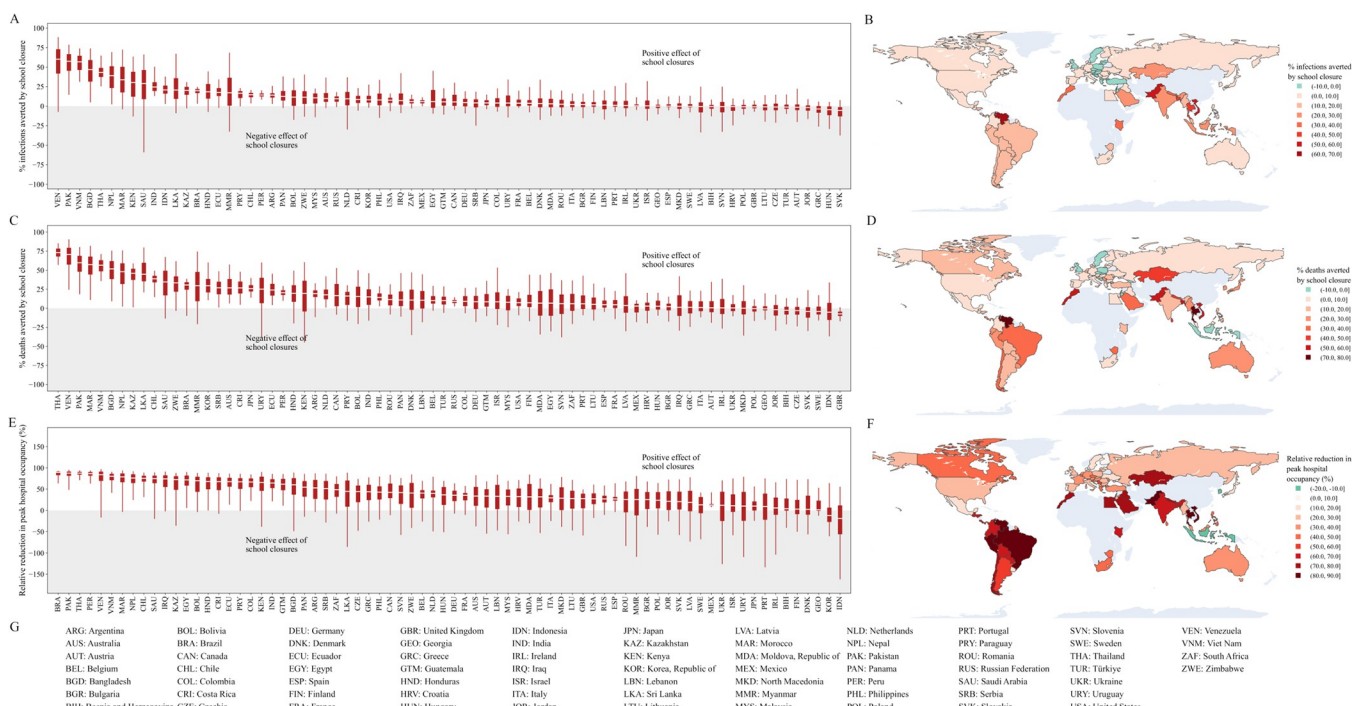

**Fig 1. Relative impact of school closures on SARS-CoV-2 infections, deaths and peak hospital occupancy pressure.** Results are presented as relative percentage reductions over the period 1 January 2020 to 31 December 2022 in SARS-CoV-2 infections (panels A and B), COVID-19-related deaths (C, D), and peak hospital occupancy pressure (E, F). The counterfactual "schools open" scenario was used as the reference for relative reduction calculations. In panels A, C, and E, estimates are presented as medians (horizontal lines), interquartile ranges (boxes), and 95% central credible intervals (vertical lines), and countries are listed in descending order from left to right, based on the estimated median effect for each disease indicator. The maps in panels B, D, and F present median estimates of relative percentage reductions, with negative values indicating configurations where school closures are estimated to have had an adverse impact on the considered indicator (based on the median estimate). Panel G presents the list of analysed countries along with their corresponding ISO3 codes. The maps were generated with *plotly* (v5.14.1) using embedded geometric data derived from the Natural Earth data set.

type" category for parsimony. We introduced differential levels of transmissibility, hospitalisation risk and death risk by strain. We assumed that preexisting immunity (both infection- and vaccine-induced) was only partially protective against the Delta and Omicron strains. The assumptions and sources of evidence used to inform strain-specific parameters are detailed in the S1 Appendix (Section A.1.4.6).

We assumed heterogeneous mixing between our modelled age groups to account for the assortative nature of social interactions by age [17]. In particular, we used age-specific contact matrices extracted from the *conmat* R package for all countries in the base-case analyses [18] and for 25 countries with available estimates from Mistry and colleagues in sensitivity analyses [19]. Both sources provide country-specific estimates for interaction rates between age groups disaggregated by the following settings: household, school, workplace, and other locations (i.e., general community). We then adjusted rates of interactions over time (dynamic mixing) to account for changes in behaviour and mobility. In particular, we used country-specific Google mobility profiles to modify the "workplace" and "other locations" components of the mixing matrices, and the UNESCO school closure database to modify the "school" component.

We further adjusted the risk of SARS-CoV-2 transmission using a time-variant random process, making the model semi-mechanistic. This random process reflected additional variations in transmission risk beyond those explicitly captured by model inputs, such as vaccination, dynamic mobility, and new variant emergence not captured by strain stratification. The process is expected to capture changes in individual behaviours such as physical distancing or

improved hygiene, as well as changes in social contact patterns due to interventions not already captured by Google mobility data and school closures. We therefore allowed for random perturbations to the risk of transmission over time, although these perturbations were constrained through autocorrelation to prevent unrealistic fluctuations within short timeframes. We used a random walk with Gaussian update to model this random process, as described in S1 Appendix Section A.1.2.6.

Finally, we combined the incidence estimates obtained from our transmission model with age-, vaccination-, and strain-specific risks of hospitalisation and deaths to compute COVID-19–related hospital occupancy pressure and deaths over time. S1 Appendix Section A.1.3.1 describes our hospital occupancy pressure indicator in detail, a quantity expected to vary proportionately with hospital occupancy over time. Note that this indicator was used in order to make comparisons between scenarios, such that one should interpret the relative differences between scenarios rather than the absolute values of the indicator. The infection fatality rate (IFR) was allowed to vary by country to account for differences in COVID-19 death definition and reporting standards. In particular, age-specific death rates were multiplied by a single scaling factor calibrated independently for each country. This approach maintains a consistent IFR age profile across countries, allowing for meaningful comparisons while accommodating country-specific variations in overall mortality rates. It also reduces analysis complexity by requiring only one mortality-related calibrated parameter per country.

We implemented the model in Python using our team's *summer* package [20], which enabled highly expressive domain-specific programming and accelerated computation through Google's *jax* library. The code used to build, run, and calibrate our model is publicly available at GitHub [21].

## Model calibration and counterfactual scenario

For each included country, we first fitted the model to local observations of reported COVID-19 deaths over time and SARS-CoV-2 seroprevalence estimates (where available). During the fitting process, the dynamic adjustment to population mixing accounted for periods of school closure informed by UNESCO data, constituting the "historical scenario". Model calibration was achieved through a Bayesian approach to account for uncertainty around key epidemiological parameters and the data used for model fitting. We employed an adaptive differential evolution Metropolis algorithm to facilitate efficient sampling of the parameter space. For each analysis, we used 8 Metropolis chains, each running for 35,000 iterations. We discarded the first 25,000 iterations of each chain as burn-in before collecting samples for analysis. Sampling convergence was assessed through trace visual inspection and the Gelman-Rubin statistic ($\hat{R} < 1.1$). S1 Appendix Section A.3 describes our calibration approach in details, along with the prior distributions used for all calibrated parameters and the model likelihood calculation. Our calibration approach resulted in a comprehensive exploration of the parameter space and identification of numerous configurations capable of replicating real-world observations. Following calibration under the historical scenario, the model simulated a "counterfactual scenario" in which schools would have remained fully open throughout the pandemic (outside of the usual academic break periods). This involved modifying the dynamic adjustment of population mixing such that rates of contact at schools are maintained at baseline levels during periods in which school closures occurred. The modifications applied to the mixing matrices to capture school closures are presented in detail in S1 Appendix Section A.1.2.5. We compared the modelled historical epidemic to our counterfactual scenarios by considering the relative differences regarding cumulative SARS-CoV-2 infections, total COVID-19 deaths, and peak hospital occupancy pressure.

### Uncertainty around social mixing assumptions and sensitivity analyses

We anticipated that model outputs would be sensitive to assumptions concerning social mixing and the impact of school closures on interpersonal interactions. We approached this issue from various perspectives. First, we introduced uncertainty in the contribution of school contacts to overall mixing. Specifically, we employed a multiplier to adjust the school component of the mixing matrices, allowing that school contact rates could contribute 80% to 120% of the values provided by the *conmat* package (base-case) or Mistry and colleagues (sensitivity analyses). Next, the UNESCO school closure database included a classification termed "Partially open" for which the exact fraction of students attending in person was unknown. We allowed this fraction to vary between 10% and 50%. The 2 uncertainty parameters mentioned above were included as calibrated parameters associated with uniform priors (see Section A.3.1 and Table B in S1 Appendix).

In addition to the uncertainty in our input parameters, we considered sensitivity analyses of parameters and assumptions most relevant to the effects of school closures, selected for their direct impact on model outcomes and based on the expert judgement of authors RR, JMT, ESM, and AEH. Specifically, we focused on assumptions regarding social interactions or the effect of school closures on contact matrices. In the base-case configuration, we assumed that school closures exclusively affected the school component of the mixing matrix, an assumption supported by a previous study in the context of influenza [22]. Although it is plausible that school closures led to increased social mixing in some settings, such as households, it is unclear whether these increases translated into higher transmission rates. Specifically, higher rates of contact within households might not lead to increased household transmission if the initial level of household interactions already approaches saturation [23]. In a first sensitivity analysis (termed SA1), we considered an alternative assumption under which effective contact rates within households were increased during school closure periods. Specifically, we assumed that each individual had 20% more effective contacts within households when schools were fully closed. In an additional sensitivity analysis (termed SA2), we removed the contribution of the Google mobility data from the modelled social mixing, thus exclusively relying on the non-mechanistic component (i.e., random process) to capture mobility changes in locations other than households and schools. Finally, our sensitivity analyses considering alternative contact matrices sourced from Mistry and colleagues (termed SA3) further explored the sensitivity of our findings to changes to social mixing parameterisation.

## Results

Based on our median estimates of intervention effects over the period 1 January 2020 to 31 December 2022, school closures were associated with a beneficial effect on infections in 58 (78%) countries, on deaths in 65 (87%) countries, and on peak hospital occupancy pressure in 72 (97%) countries (Fig 1). However, the 95% credible intervals (95 CIs) around our estimates for the 3 disease indicators were wide for most countries, such that even the direction of the effect was difficult to evaluate with certainty. Only 11 (14%) countries were associated with a 95 CI entirely consistent with a positive effect of school closures on infections, whereas 20 (27%) countries had a 95 CI suggesting a positive effect on deaths, and 13 (17%) had a 95 CI suggesting a positive effect on peak hospital occupancy pressure. Fig 1 provides a geographic representation of the median effects of school closure on infections, deaths, and peak hospital occupancy pressure. We estimated that school closures significantly averted infections in all analysed countries of Southeast Asia and the Indian subcontinent. In contrast, the USA and 25 of the 30 European countries analysed were associated with small or negative estimated impacts of school closures on infections and deaths, with estimated median reductions of less

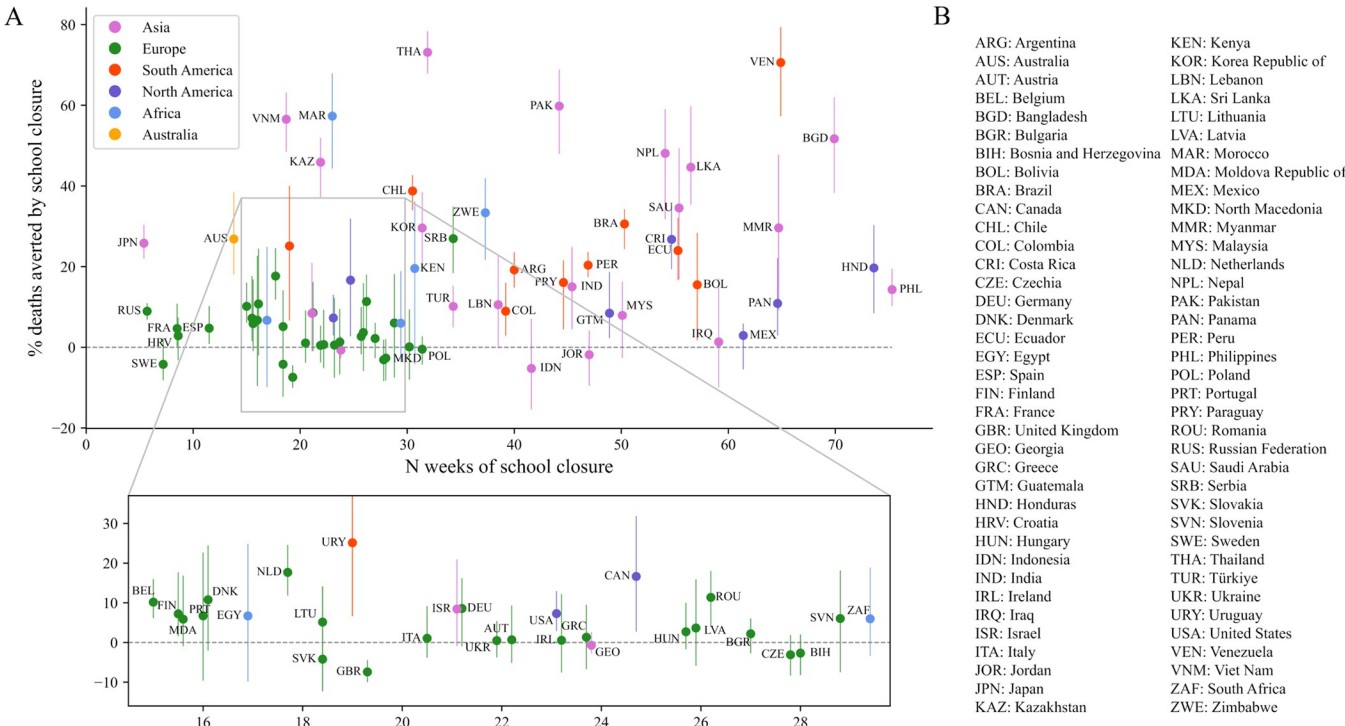

**Fig 2. Relationship between the estimated effect of school closures on COVID-19 deaths and the total duration of school closures.** In Panel A, relative reductions of COVID-19 mortality over the period 1 January 2020 to 31 December 2022 are presented as medians (dots) and interquartile ranges (bars), with colours indicating the continents to which the countries belong. The total number of closure weeks was obtained by adding the number of weeks fully closed, and the number of weeks partially closed multiplied by 0.3 (our midpoint estimate for assumed attendance fraction during partial closures). Panel B presents the list of analysed countries along with their corresponding ISO3 codes.

than 10% for both indicators. In all countries of Central and South America, our model suggested major reductions of peak hospital occupancy pressure due to school closures. The intervention was also estimated to have had a positive impact on infections, deaths, and peak hospital occupancy pressure in the 5 African countries analysed. Fig 2 indicates that the estimated effect of school closures on deaths was not clearly correlated with the total closure duration. Notably, the 2 countries with cumulative closure duration exceeding 70 weeks (the Philippines and Honduras) only achieved a modest reduction in COVID-19 mortality (median reduction: 14% and 20%, respectively). In contrast, Vietnam and Morocco achieved mortality reductions of over 40% with school closures lasting less than 25 weeks, while Japan was estimated to have reduced COVID-19 mortality by 26% (median estimate) with only 5 weeks of school closures. We present detailed outputs for 3 selected countries in Fig 3 (Morocco), Fig 4 (Indonesia), and Fig 5 (United Kingdom), and for all analysed countries in S1 Appendix Section B.3. We also provide an online interactive tool for detailed exploration of all the analysis outputs [24]. The 3 countries presented in the main text were selected for their different epidemic profiles and the diverging conclusions obtained. In Morocco, our model suggested that school closures were instrumental in preventing a huge first epidemic wave in 2020, which alone would have caused significantly more infections and deaths than the total estimated over the whole period 2020 to 2022 under the historical scenario (Fig 3). Similar observations were made in other countries including Bangladesh, Bolivia, Honduras, and India (S1 Appendix and Section B.3). In Indonesia, while our analysis suggested that school closures prevented a large epidemic wave in 2020, it also showed that the large Delta variant wave observed in the

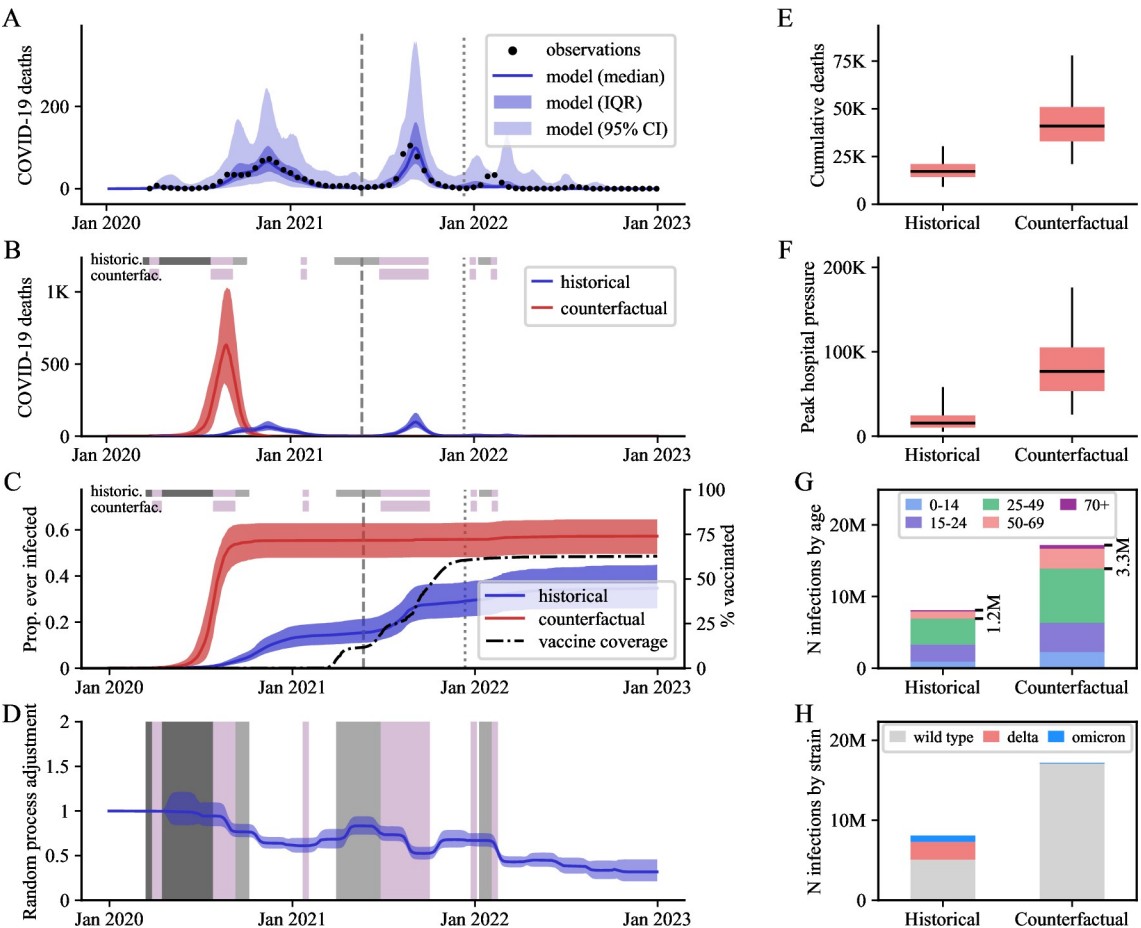

**Fig 3. Detailed outputs for Morocco, as a representative country where closures significantly reduced COVID-19 infections, hospitalisations, and deaths.** (A) Modelled COVID-19 deaths over time against observations. (B, C) Scenario comparison over time for COVID-19 deaths (B) and proportion infected (C). (D) Estimated transmission adjustment over time (random process) compared with school closure timelines. (E, F) Scenario comparison for cumulative deaths (E) and peak COVID-19–related hospital occupancy pressure (F) over the period 1 January 2020 to 31 December 2022. (G, H) Cumulative SARS-CoV-2 infections by age (G, age expressed in years) and strain (H) over the same period. Uncertainty is represented with dark shaded areas (IQR, interquartile ranges) in Panels A–D, and light shaded areas (95% central credible intervals) in Panel A. Boxplots E and F present estimates as medians (horizontal lines), interquartile ranges (boxes), and 95% central credible intervals (vertical lines). School closure statuses are illustrated with the horizontal coloured bands in Panels B–D (dark grey: fully closed for COVID-19, light grey: partially closed due to COVID-19, purple: academic break). In Panels A–C, the vertical lines indicate the emergence dates of the Delta variant (dashed line) and the Omicron variant (dotted line) in the country according to the GISAID database. K, thousands; M, millions; Jan, January.

second half of 2021 may not have occurred under the counterfactual "schools open" scenario (Fig 4). This is explained by the much larger population immunity built by the time Delta emerged under the counterfactual scenario as compared to the historical scenario (Fig 4, Panel C). This observation, combined with the increased severity of the Delta variant compared to ancestral strains, explain the worsening of cumulative COVID-19 mortality and peak hospital occupancy pressure caused by school closures in Indonesia according to our model. We explored whether more stringent school closures during the Delta wave could have reduced COVID-19 mortality in Indonesia. With schools only partially closed at the time, we considered a counterfactual scenario with full closures throughout 2021. Our estimates suggest that this intervention alone would have been insufficient to reverse the trajectory of Indonesia's Delta wave, resulting in only a relatively small reduction in COVID-19 burden (S1 Appendix

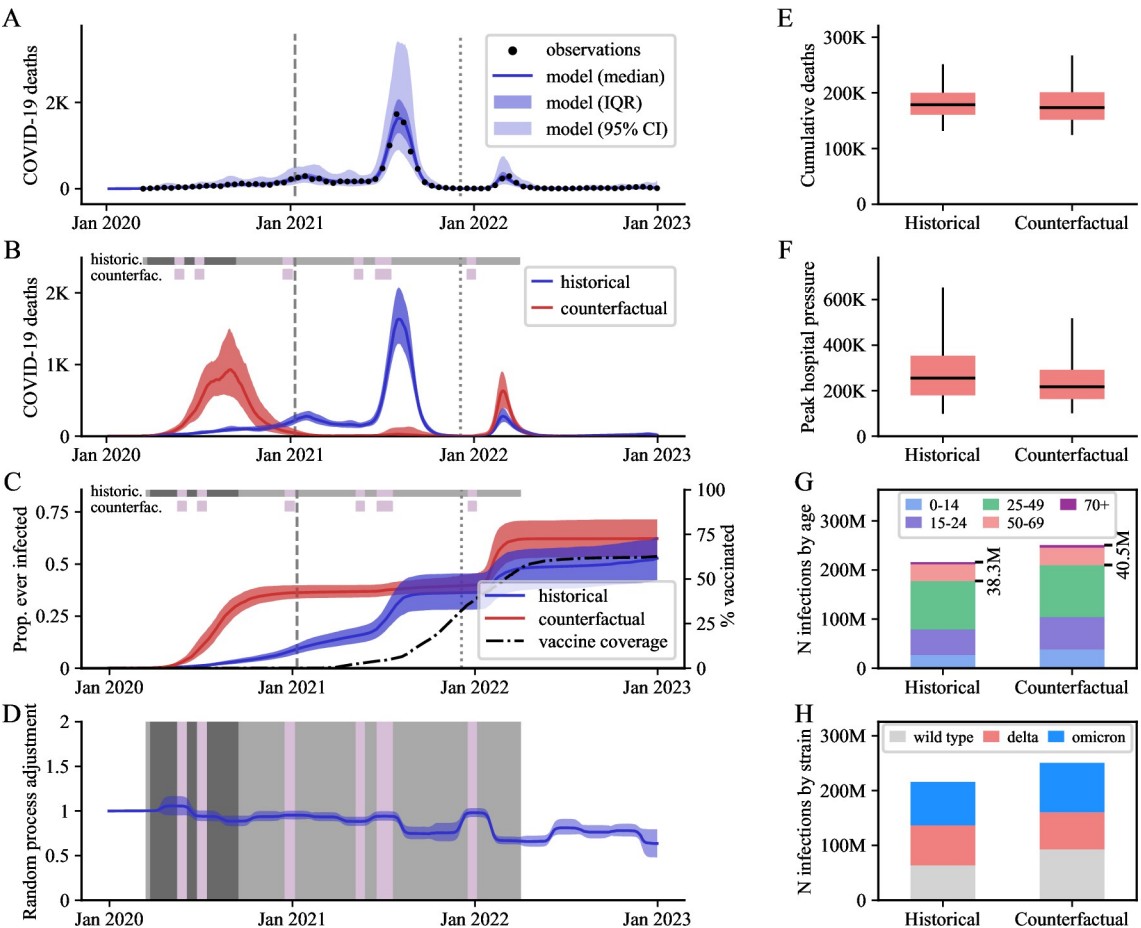

**Fig 4. Detailed outputs for Indonesia, as a representative country where closures may have increased COVID-19 mortality due to an exacerbated Delta wave.** (A) Modelled COVID-19 deaths over time against observations. (B, C) Scenario comparison over time for COVID-19 deaths (B) and proportion infected (C). (D) Estimated transmission adjustment over time (random process) compared with school closure timelines. (E, F) Scenario comparison for cumulative deaths (E) and peak COVID-19–related hospital occupancy pressure (F) over the period 1 January 2020 to 31 December 2022. (G, H) Cumulative SARS-CoV-2 infections by age (G, age expressed in years) and strain (H) over the same period. Uncertainty is represented with dark shaded areas (IQR, interquartile ranges) in Panels A–D and light shaded areas (95% central credible intervals) in Panel A. Boxplots E and F present estimates as medians (horizontal lines), interquartile ranges (boxes), and 95% central credible intervals (vertical lines). School closure statuses are illustrated with the horizontal coloured bands in Panels B–D (dark grey: fully closed for COVID-19, light grey: partially closed due to COVID-19, purple: academic break). In Panels A–C, the vertical lines indicate the emergence dates of the Delta variant (dashed line) and the Omicron variant (dotted line) in the country according to the GISAID database. K, thousands; M, millions; Jan, January.

Section B.2.7). In the UK, while we estimated that school closures reduced the total number of infections, we also found that they may have increased the number of infections in the over-50-years-old population (Fig 5, Panel G). Indeed, more infections were required in the older population to compensate for the reduced transmission in the younger age groups, ultimately achieving a comparable level of population immunity. The shift in the age distribution of infections combined with the higher IFR in older age groups explain our finding that school closures likely increased overall COVID-19 mortality in the UK.

The estimated transmission adjustment (random process) over time is presented for the 3 highlighted countries in Panel D of Figs 3–5 and for all analysed countries in S1 Appendix. Our model consistently captured local epidemic dynamics without requiring abrupt changes in the random process. Notably, we found no systematic shifts coinciding with school closure

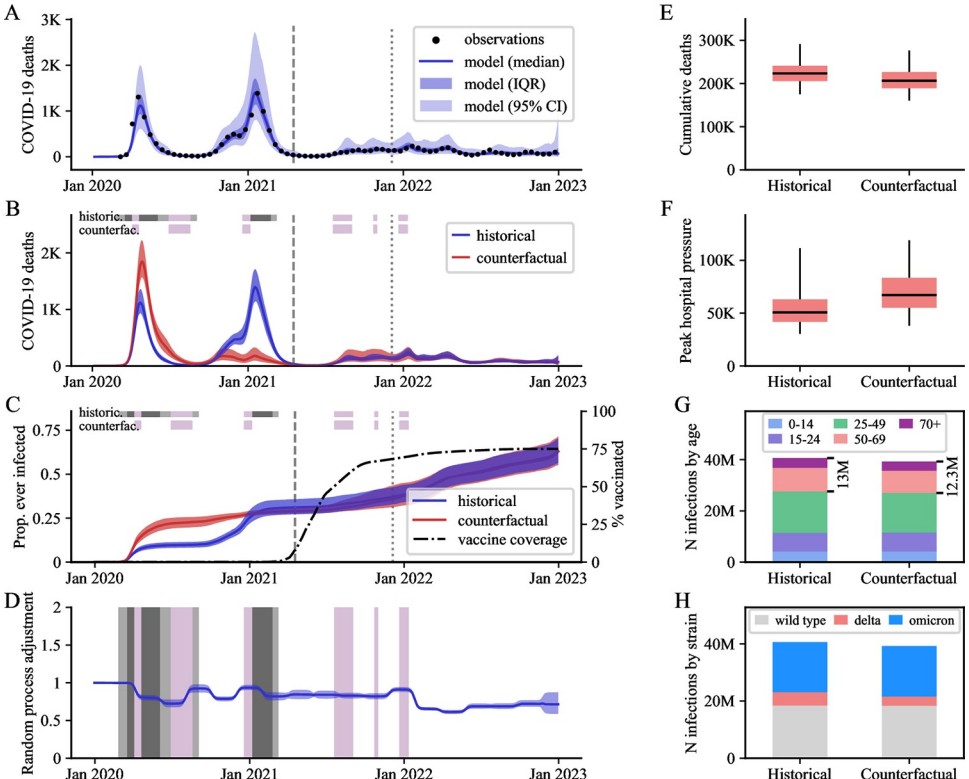

**Fig 5. Detailed outputs for the United Kingdom, as a representative country where closures may have increased COVID-19 mortality due to a shift in the age distribution of infections.** (A) Modelled COVID-19 deaths over time against observations. (B, C) Scenario comparison over time for COVID-19 deaths (B) and proportion infected (C). (D) Estimated transmission adjustment over time (random process) compared with school closure timelines. (E, F) Scenario comparison for cumulative deaths (E) and peak COVID-19–related hospital occupancy pressure (F) over the period 1 January 2020 to 31 December 2022. (G, H) Cumulative SARS-CoV-2 infections by age (G, age expressed in years) and strain (H) over the same period. Uncertainty is represented with dark shaded areas (IQR, interquartile ranges) in Panels A–D, and light shaded areas (95% central credible intervals) in Panel A. Boxplots E and F present estimates as medians (horizontal lines), interquartile ranges (boxes), and 95% central credible intervals (vertical lines). School closure statuses are illustrated with the horizontal coloured bands in Panels B–D (dark grey: fully closed for COVID-19, light grey: partially closed due to COVID-19, purple: academic break). In Panels A–C, the vertical lines indicate the emergence dates of the Delta variant (dashed line) and the Omicron variant (dotted line) in the country according to the GISAID database. K, thousands; M, millions; Jan, January.

or reopening times, suggesting that the transmission changes induced by school closures were adequately captured by our approach based on mixing matrix modifications. However, we observed a gradual decline in the random process in some countries (e.g., Morocco, Fig 3), suggesting the presence of factors not explicitly captured by the model that may have contributed to reduced transmission over time (e.g., increased awareness, improved hygiene).

Assuming increased household transmission during periods of school closures (SA1) was associated with lower estimated benefits from school closures in most countries, regardless of the indicator considered (infections, deaths, or peak hospital occupancy pressure; Figs N and S–U in S1 Appendix). However, our main conclusions remained broadly unchanged regardless of the mixing assumption implemented. The 2 configurations in which Google Mobility data were used to inform the modelled dynamics (Base-case and SA1) were associated with higher a posteriori likelihood values in most countries compared to the configuration where these data were excluded (SA2), suggesting that incorporating Google Mobility data helped achieve more realistic model fits (Fig W in S1 Appendix). In particular, we found that the

time-variant random process, which reflects transmission changes not explicitly captured by the model inputs, required less variability under the configurations where mobility data was incorporated in multiple countries (Fig X in S1 Appendix).

Using an alternative source of contact matrix estimates (SA3) did not lead to significant changes in the estimated school closure effects in the 25 countries where we were able to conduct this sensitivity analysis (Figs R–U in S1 Appendix). Similarly, varying the number of compartments used to represent the incubation and active disease periods produced model outputs that were virtually identical to those of the base-case analyses (Figs Y, Z, AA, and AB in S1 Appendix).

To investigate potential sources of heterogeneity in the effects of school closures on COVID-19 dynamics, we conducted a post hoc analysis examining the relationships between our findings and key population characteristics (see S1 Appendix Section B.1). This analysis revealed a negative association between the proportion of individuals aged 70 and older and the effectiveness of school closures in reducing infections and deaths. Additionally, the analysis suggested that school closures may have been more beneficial in settings where less stringent public health and social measures other than school closures were in place during the closure periods, although this association did not reach statistical significance.

## Discussion

We conducted a systematic analysis of the impact of school closures on the dynamics of COVID-19 across 74 countries with markedly different pandemic experiences. Employing a purpose-built semi-mechanistic model, we explored the intricate relationship between school closures and SARS-CoV-2 transmission, mortality, and hospital occupancy pressure. We integrated various data sources to inform and calibrate our model, and considered a counterfactual scenario to estimate the potential outcomes if schools had remained open.

Our findings revealed nuanced and context-specific effects of school closures. While a large majority of countries likely experienced significant overall reductions of infections, COVID-19 deaths and hospital occupancy pressure due to school closures, our model also highlights the potential for negative epidemiological impacts from this intervention. We estimated that, in many settings, school closures were highly effective in mitigating the impact of COVID-19 during the earliest stages of the pandemic. For example, in countries such as Morocco, Bolivia, and India, the early reductions were so significant that a single counterfactual wave occurring with schools open would have resulted in more infections and deaths than the total observed during the first 3 years of the pandemic under the historical scenario. We further found that school closures likely reduced peak hospital occupancy pressure in 72 out of the 74 analysed countries, according to our median estimates. However, in countries such as Indonesia, Argentina, and Lebanon, school closures prevented one epidemic wave but may have contributed to the exacerbation of another due to the lack of population-level immunity when the more severe Delta variant emerged.

The question of whether school closures were a good policy still lacks a straightforward answer. Even for a single country, the estimated epidemiological effects may differ for various indicators, as illustrated with Indonesia where closures prevented infections but increased COVID-19 mortality. Moreover, our estimates were associated with substantial uncertainty, stemming from challenges in accurately characterising certain infection features and the inherent difficulty in precisely predicting individuals' behaviour in the counterfactual scenario where schools remained open. Furthermore, ascertaining the policy's effects on COVID-19 dynamics represents only one facet of the broader societal impact, considering that school closures had profound consequences on other aspects including mental health, education, and economy.

Despite the uncertainties, our study identified critical mechanisms that must be considered in future policy decisions. While school closures were mostly estimated to have had positive effects on COVID-19 epidemics, delaying epidemic waves using non-pharmaceutical interventions may not always be advantageous given the unpredictable nature of emerging variants. Additionally, school closures may lead to a shift in the age distribution of infections, potentially impacting older age groups more severely as observed in our projections for the UK. Such changes in the age-distribution of infections were also reported in a previous study investigating the impact of school closures in Germany [25]. These insights underscore the importance of a nuanced approach to the implementation of public health measures. However, it is crucial to recognise the limitations of hindsight when assessing the decisions made during the early days of the pandemic. Little was known about COVID-19 at that time, including age-specific characteristics, future variants' severity and transmissibility, postinfection immunity, and vaccine effects. Our intention is not to criticise past policies but to shed light on key mechanisms of intervention effectiveness and highlight potentially under-recognised factors that should be considered at decision time in the future.

In many instances, governments have employed school closures as a reactive measure in response to escalating epidemics or localised outbreaks. However, there remains considerable uncertainty regarding the optimal trigger for implementing such policies, whether based on hospital capacity thresholds, case numbers, or other indicators. While we did not evaluate specific triggers, future research could explore various reactive strategies, comparing different indicators and threshold levels to better inform decision-making. Accurate testing and contact tracing procedures may also enable more effective, targeted closures at the school or even class level. Our finding that longer school closure durations did not consistently lead to better epidemiological outcomes suggests that carefully designed strategic policy approaches may potentially achieve more favourable results with less stringent restrictions.

Previous studies investigating the epidemiological impact of school closures on COVID-19 mostly focused on the instantaneous effects on transmission through estimations of reproduction number or growth rate. Most studies focused on specific countries such as Canada [26], France [27], Germany [25], and China [28], or applied conceptual models to theoretical settings [29]. Despite the variation in scope and settings, the collective evidence indicates that school closures tend to reduce SARS-CoV-2 transmission [27,28,30–32], which is consistent with our findings regarding the short-term effects of the intervention. Our approach differs fundamentally from previous works from at least 2 aspects. First, instead of trying to estimate the changes in reproduction number induced by school closures, we captured transmission changes explicitly by modifying the school component of country-specific mixing matrices over time. Second, we considered the comprehensive impact of school closures over prolonged periods spanning multiple epidemic waves. To our knowledge, our study stands as the first to assess the effects of various factors such as the emergence of different viral strains, vaccination, and infection-induced immunity in relation to school closures.

Strengths of our study include our state-of-the-art methods for model coding, running, and calibration, using only free and open-source software, and making code and model outputs publicly available for transparency and reproducibility [21,24]. The use of advanced computing techniques facilitated the creation and enabled the calibration of a sophisticated model capturing critical factors for understanding COVID-19 epidemiology across 74 countries. Key components of our model were tested through sensitivity analyses, confirming the robustness of our findings despite variations in assumptions. Finally, our semi-mechanistic approach drew inspiration from a previously published model successfully applied to COVID-19 vaccination [16]. Notably, one of our enhancements over this prior study lies in the incorporation of not just COVID-19 death time series for model calibration but also seroprevalence estimates

where available. This substantial improvement allows for a more accurate understanding of the true size of the epidemic, encompassing both infections and deaths.

Despite our efforts, limitations persist. The counterfactual scenario inherently involves uncertainty, as we cannot precisely determine what would have happened under a configuration that did not occur in reality. In particular, behavioural changes in response to local disease pressure remain poorly characterised and, therefore, challenging to incorporate in a model. Moreover, uncertainties persist around the data used to inform our analyses. First, the GISAID database provides the date of first reported case for each variant but this may differ from the time when the variant effectively started to spread locally. Second, the "Partially closed" category reported by the UNESCO school closure database lacks specificity and may need to be interpreted differently in different countries, such that we employed a broad uncertainty range around the modelled school attendance fraction during these periods. Finally, due to the absence of contact survey data for all included countries, the mixing matrices capturing age- and location-specific interactions were derived from extrapolation. We partially addressed this limitation by injecting uncertainty in the relative contribution of school contacts to overall mixing, as this quantity was anticipated to play a significant role in our analysis. We also tested alternative contact matrices in sensitivity analyses, which resulted in only marginal changes to our estimates.

Despite our efforts to incorporate numerous factors relevant to SARS-CoV-2 transmission, our model did not account for every mechanism influencing disease spread and evolution over time. Specifically, changes in individual behaviours, such as mask-wearing and social distancing practices, were not explicitly modelled, beyond what could be inferred from mobility data. However, the use of an automatically calibrated random transmission adjustment helped to account for unmodeled factors. Importantly, we did not observe abrupt changes in this random adjustment, suggesting that the factors explicitly captured by our model were sufficient to explain most of the observed variations in COVID-19 dynamics. Future work may expand on our analysis by considering a more limited set of countries or regions with detailed information on individual behaviours and specific control policies, providing more precise insights into the effects of school closures.

While we aimed to include a comprehensive range of countries in our analysis, certain regions were underrepresented due to data limitations. The small number of included African countries was primarily explained by the lack of data on variants emergence times from the GISAID database. We attempted to include more African countries after manually collecting variants data from alternative sources. Unfortunately, the model did not produce interpretable results due to the relatively low number of reported COVID-19 deaths over time, which was the main indicator used for model calibration, and the lack of seroprevalence data. Additionally, we could not provide estimates for China due to the unavailability of Google mobility data.

Future research could expand on our findings. Collaborations with social science experts could provide a holistic understanding of the broader impacts of school closures, beyond COVID-19–related indicators. Subnational analyses would offer more granularity, considering that school closures often occurred non-uniformly within countries. However, the lack of publicly available subnational data for key factors, such as school closure timelines and model calibration inputs, made it unfeasible to incorporate this level of detail in our analysis across numerous countries. Finally, the effects of the policy could be further explored using agent-based models which may better capture the dynamic effects of mixing changes during closures.

In conclusion, our study sheds light on the intricate relationship between school closures and the dynamics of COVID-19. The findings emphasise the complexity of assessing the

overall impact of this intervention, given the variability across countries and the multitude of factors involved. Moving forward, a nuanced and context-specific approach is essential in making decisions about school closures and other non-pharmaceutical interventions, considering the evolving nature of the pandemic and the multifaceted consequences on societies worldwide.

## Supporting information

**S1 Appendix. Methodological details and additional results.**
(PDF)

## Acknowledgments

The Epidemiological Modelling Unit at the School of Public Health and Preventive Medicine (EMU) provided modelling to countries of the Asia-Pacific through a series of contracts with the World Health Organization Western Pacific and South East Asia Regional Offices over the course of the pandemic, through which much of the software development underpinning this analysis was undertaken.

## Author Contributions

**Conceptualization:** Romain Ragonnet, David S. Shipman, Michael T. Meehan, James M. Trauer.

**Data curation:** Romain Ragonnet.

**Formal analysis:** Romain Ragonnet, Michael T. Meehan, Guillaume Briffoteaux, Daniel Tuyttens.

**Funding acquisition:** James M. Trauer.

**Investigation:** Romain Ragonnet, Angus E. Hughes, Guillaume Briffoteaux.

**Methodology:** Romain Ragonnet, Michael T. Meehan, Alec S. Henderson, Guillaume Briffoteaux, Nouredine Melab, Daniel Tuyttens, Emma S. McBryde, James M. Trauer.

**Project administration:** Romain Ragonnet, Angus E. Hughes.

**Resources:** Angus E. Hughes.

**Software:** Romain Ragonnet, David S. Shipman, Alec S. Henderson, Nouredine Melab, James M. Trauer.

**Supervision:** James M. Trauer.

**Validation:** Romain Ragonnet, Angus E. Hughes, Emma S. McBryde.

**Visualization:** Romain Ragonnet, David S. Shipman.

**Writing – original draft:** Romain Ragonnet.

**Writing – review & editing:** Romain Ragonnet, Angus E. Hughes, David S. Shipman, Michael T. Meehan, Alec S. Henderson, Guillaume Briffoteaux, Nouredine Melab, Daniel Tuyttens, Emma S. McBryde, James M. Trauer.

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
