## [Editor Report · Decision Letter 0]

26 Jun 2024

Dear Dr Ragonnet, 

Thank you for submitting your manuscript entitled "Estimating the impact of school closures on the COVID-19 dynamics in 74 countries: a modelling analysis" for consideration by PLOS Medicine.

Your manuscript has now been evaluated by the PLOS Medicine editorial staff and I am writing to let you know that we would like to send your submission out for external peer review.

Please re-submit your manuscript within two working days, i.e. by Jun 28 2024.

Feel free to email me at atosun@plos.org or us at plosmedicine@plos.org if you have any queries relating to your submission.

Kind regards,

Alexandra Tosun, PhD

Associate Editor

PLOS Medicine

---

## [Decision Letter · Decision Letter 1]

15 Aug 2024

Dear Dr Ragonnet,

Many thanks for submitting your manuscript "Estimating the impact of school closures on the COVID-19 dynamics in 74 countries: a modelling analysis" (PMEDICINE-D-24-02032R1) to PLOS Medicine. The paper has been reviewed by subject experts and a statistician; their comments are included below and can also be accessed here: [LINK]

As you will see, the reviewers were positive about the manuscript, but raised a similar set of issues including but not limited to model specification and choice of input data. After discussing the paper with the editorial team and an academic editor with relevant expertise, I'm pleased to invite you to revise the paper in response to the reviewers' comments. We plan to send the revised paper to some or all of the original reviewers, and we cannot provide any guarantees at this stage regarding publication.

We ask that you submit your revision by Sep 05 2024. However, if this deadline is not feasible, please contact me by email, and we can discuss a suitable alternative.

Don't hesitate to contact me directly with any questions (atosun@plos.org). 

Best regards, 

Alexandra 

Alexandra Tosun, PhD 

Associate Editor

PLOS Medicine

atosun@plos.org

Comments from the academic editor:

It is a strong manuscript, with some interesting findings. The comments are helpful, and it seems that the manuscript has potential to be an important addition to the scientific literature about a contentious and under researched issue (effect of school closures).

A couple of suggestions:

1) Some figures, particularly 3,4 and 5, would benefit from simplification and avoidance of trying to present too much data at once.

2) More emphasis in the discussion on the lack of data from Africa. Having lived and worked in Africa during the COVID-19 pandemic, I am acutely aware that many countries implemented long school closures based on what was happening elsewhere in the world, and probably to the considerable detriment of children's well-being and education. It is a shame that more African countries did not have data to include.

3) I also note that there seemed to be no data available for China to support inclusion - good if the authors can comment on this.

4) Although the presentation of relative risks is helpful, I would probably have liked to see absolute numerical magnitudes of the effects on outcomes under counterfactual scenarios. This would be very helpful for public health planning.

Comments from the reviewers: 

Reviewer #1: I commend the authors for such a well-written, novel, and well-conceived manuscript. The topic will be of interest to readers. While other researchers have attempted to clarify the effect of school closures on COVID-19 outcomes, doing so remains of importance and this paper offers a novel approach to this question via 1) inclusion of a remarkable 74 countries in the analysis, and 2) exploration of the effect of school closures over a longer period of time, 2020 to 2022. Additionally, the paper demonstrates some novel findings, that are presented in an intuitive way. First, the authors find that, in some countries, the avoidance of a surge in the wildtype strain from school closures led to lower immunity and thus higher cases and more hospitalizations during the delta wave. Second, the authors demonstrate that school closures in some cases shifted the age distribution of infections to older individuals, resulting in an overall increase in deaths. 

Methodologically, the work is sound. The authors have previously published this model in other settings and are highly transparent about model functionality through a well-written supplement and sharing of the code in a public GitHub repo. They add additional components to this model beyond their previously published work via a semi-mechanistic component and model calibration with seroprevalence data and death data.

There are, as to be expected, some limitations of this work. Notably, there is substantial uncertainty surrounding childhood behavior that cannot be captured by the UNESCO database. In the US, where this reviewer is from, the "partial reopening" period was clustered by geographies (e.g., some states opened far sooner than others), and in the "fully reopen" period, many districts offered the choice to parents to continue to keep their children home for remote instruction. Adherence to social distancing was highly heterogeneous at the sub-national level. However, the authors note both these limitations in the discussion and account well for uncertainty, including through sensitivity analyses. Other methodological limitations I observed include the assumption that individuals cannot be infected twice with the same strain, imperfect immunity seems to be constant rather than reflect waning, vaccination patterns (saturating the elderly before moving to younger age groups) may be unrealistic, and hospital pressure is based off values from the Netherlands (although adjusted by country for death rate). Nevertheless, I find these to be reasonable assumptions to make within an already sophisticated and nuanced model framework.

In sum, it was a delight to review this manuscript and am pleased to have no substantial comments on this work. I can only offer small suggestions and observations:

1. Because the time period of the study was only introduced once in the intro, it would be helpful to add within results and figure legends that the estimated relative reductions are over the period January 1, 2020 to December 31, 2022.

2. I noted that the Github links on page 15 of the supplement did not work for me. I was able to access what I believe are the correct links, with the difference in url's being "master" vs. "main"

3. I spent a lot of time trying to interpret the vaccination-related terms in A.5 equations in the supplement. First, I could not find the definition of phi because it came after the equations. Then, I saw the definition and understood phi, but continued to have trouble with its interpretation, including for the subscript v=0, as in S_v=0. Perhaps it may help readers to have phi introduced in the text above the equations with an additional sentence surrounding the product of phi with w(t) and the state variable conditioned on v=0.

Reviewer #2: Authors of this paper analyze a mass of data collected during the COVID-19 pandemic describing the epidemic dynamics, vaccination coverage, dynamics of variants, mobility, timing and nature of school closures in 71 countries to ascertain the impact of school closure (as measured by the proportion of cases, hospitalizations and deaths averted thanks to the intervention) during the pandemic. They find that in most countries school closure had a beneficial impact but that in a few countries, counterfactual scenarios without school closure led to smaller epidemic burden.

I find that this is an interesting attempt to quantify the impact of school closure, looking at data from a large number of countries. However, I have major comments about the methods, the results, their interpretation and validity.

1) Is school closure impact inferred from the data or is it mostly a byproduct of modelling assumptions and model structure? In general, in studies that aim to evaluate the impact of interventions on transmission, the transmission rate is modelled as a function of the interventions being implemented at a given time, for example the modelled transmission rate is reduced by a factor beta_school when schools are closed. The impact of school closure is then measured by estimating from the data parameter beta_school. What puzzles me with this work is that there does not seem to be parameters that are estimated from the data to quantify the impact of school closure. The only two I could find are the "uncertainty multiplier for school contact" (between 0.8 and 1.2 so always close to 1) and the "proportion of students on site during partial opening". As a result, it appears that estimates of school closure impact are largely driven by modelling assumptions, specifically equation A.3 in the Supplement describing the structure of the contact matrix that includes a matrix for schools. 

So, I expect that assumptions about the contact matrices and how they changed over time during the pandemic are driving the results. 

a. I'm not familiar with the package conmat that was used to generate these matrices, put presumably for a relatively large number of countries, no contact data are available and the contact matrix is reconstructed from data available in countries with similar profiles, which might be particularly problematic in countries with atypical demographic and cultural structures.

b. Properly characterizing how these contact matrices changed over time during the pandemic is a major challenge and there is a serious risk of model misspecification here. I didn't find it very clear how authors performed this task. I understand this was done through google mobility; but authors need to provide more details.

An important risk with the approach used is that results may be wrong if the model is misspecified (which is a serious possibility). The problem is exacerbated by the fact that there is currently no way to ascertain model adequacy (see next point). I think that this represents an important limitation of the work. For readers, it is essential to clarify what is being done here, i.e. it is not about estimating the instantaneous impact of school closure on transmission, rather, under a setting of assumptions about contact matrices and the instantaneous impact of school closure on transmission, it is to derive overall impact of school closure. Discussion needs to be extended to reflect the fact that results are highly dependent on model assumptions, in particular the modelling of the contact matrices and the important risk of misspecification, particularly in the pandemic context.

2) Model adequacy: When a simple mechanistic model (for example with 1 transmission rate when schools are open and 1 when schools are closed) presents a good fit to the epidemic time series, this offers assurances that the school closure model captures key features in the data. Here, although fits to country data are almost systematically good, such good fits are not necessarily indicative that the impact of school closures is well captured by the model. This is because other interventions are modeled with a very flexible autocorrelated process. With this approach, any mismatch between the school closure model and the data will be compensated by the autocorrelated process so that the overall fit will be good. This is problematic because it means we have no way to assess if modelling assumptions about school closure are supported by the data. 

The model is only calibrated to death counts and serological data. Overall death are very aggregated data that are unlikely to provide the granular insight necessary to capture the impact of an intervention such as school closure targeting kids. 

It seems important that authors manage to identify a signature in the data for the impact of school closure (e.g. a change in the age structure of cases, a slowing down of growth in children) and that they show that their model can reproduce this signature and that a model without school closure could not reproduce this signature. I think this is difficult to achieve in a context where a lot of other interventions were implemented.

It would also be useful to be able to compare impact derived on instantaneous R in different countries to assess consistency between estimate and see if these estimates make sense. 

3) Interpretation of results: Authors provide for each country the estimated number of cases, deaths and hospitalizations at the peak averted thanks to school closure. For example number of cases averted due to school closure go from more than 50% in Venezuela to slightly negative value for Great Britain. Currently, there is no much mechanistic insight about why we get these differences. Authors explain a bit why in some cases the impact might have been negative (because of an absence of build up of immunity in children); but this does not explain the huge heterogeneity observed between countries (and why this phenomenon only occurred in a very small subset of countries). Without good understanding of the drivers of this heterogeneity, it is very hard to ascertain whether the results across the different countries are consistent with each other or simply reflect a model that is overparametrized and produce inconsistent results from one country to the next. It is therefore essential that authors manage to better explain what drives heterogeneity in country specific estimates:

a. The duration of closure is a first natural driver. 

b. Most importantly, I would hypothesize that the impact of school closure may largely depend on the intensity of other interventions. For example, in a country where the rest of the population is in lockdown, closing schools might have only limited additional impact. In contrast, if mixing patterns in other environments are close to normal, school closure may have a more important contribution to parameter. 

c. It might also be interesting to explore how other types of school based interventions might affect results. For example, in some countries, school closure was limited but there were a lot of other school based interventions, e.g. masks wearing, repeated testing etc. Do school closures remain relevant in this context?

d. Could the population structure affect impact? For example with bigger impact in younger populations?

4) Emphasis on the negative impact of school closure: There is currently an important emphasis in the abstract, results and discussion on the potentially negative impact of school closure. However, I don't find it convincing at all. Just looking at figure 1, there is a first set of countries where school closure clearly had a positive impact; and then a second set where impact was negligeable. There are just 1-3 countries out of 71 where estimates suggest a slightly borderline negative effect so I would really question the strength of evidence on this negative effect. Yet, this weak finding is strongly emphasized and takes a big part of abstract/results/discussion.

As indicated in the previous point, it would be much better to attempt to explain heterogeneity in estimates rather than spending a large part of paper explaining patterns in 1-3 countrie out of 71.

Other comments:

Line 69 : "We introduced differential levels of transmissibility, hospitalisation risk and death risk by strain. We assumed that pre-existing immunity (both infection- and vaccine-induced) was only partially protective against the Delta and Omicron strains." Please comment on assumption made and data supporting them (a reference to Supplement is fine).

High autocorrelation in random effect to "avoid unrealistic changes over a short period of time". But in a lot of countries, very important changes occurred over short periods of time, for example at the start of lockdowns in the first wave in 2020. The autocorrelation might lead to underestimating the impact of the lockdown and overestimating that of school closure (for which sudden nature of change is presumably well captured by model. 

Reviewer #3: This manuscript presents a mathematical model used to evaluate the impact of school closure policies on the dynamics of the COVID-19 pandemic in 2020-2022 in 74 countries. The authors claim that school closures generally reduced the COVID-19 burden but had a negative impact in some countries (USA and Europe), probably attributable to a shift in the age distribution of cases. 

The article is well written and organized in general, and I appreciate that the authors made the code and data used for their analyses available on Github. 

The topic is interesting as the role of school closures in contrasting an epidemic is still not completely clear and deserves thorough investigation through a modeling approach; advances in the knowledge of the advantages and drawbacks of this non-pharmaceutical policy would surely have an impact on public health decisions. 

However, I have some perplexity on the presented modeling choices, and I also believe that the findings of this work would not represent a considerable contribution to the topic. This considered, I would not recommend this work for publication into PLOS Medicine; I would suggest considering PLOS ONE after revision of this manuscript. 

Major comments 

1. The transmission model implemented by the authors is quite complex and to a certain extent unconventional: 

- two exposed compartments are partially infectious, but they are not considered as infectious compartments 

- the random perturbation of the transmission risk, W(t), is non-standard 

- the nomenclature used for the compartments is unusual ("actively diseased" instead of "infectious") 

The authors should provide additional information to justify their modeling choices and support them by showing in the Supplementary material their appropriateness and the improvements achieved as compared to a standard compartmental model. Lacking this, it is difficult to understand whether these results would be confirmed by a standard, well-established model. 

2. The authors state "We further assume that the infectious exposed compartments (E3 and E4) are half as infectious as the active disease compartments" without motivating this choice. Is there a supporting reference? 

3. Active disease compartments are considered all equally infectious, but infectivity is not constant throughout the infectious period of an individual. 

4. Since the authors report different impact of school closures on infections/hospitalizations/deaths for different areas of the world, I believe that this study would benefit from the inclusion of details on the possible socio-demographic factors that may be involved: correlation with age structure of the population, fraction of active individuals (i.e. students or belonging to the workforce), etc. 

5. The authors observe that, for some countries, school closure may have caused an overall worsening of COVID-19 mortality by contributing to the emergence of the Delta wave. It would be interesting to include a second counterfactual scenario implementing a reactive school closure policy where closure is based on real-time monitoring of some epidemiological indicators (e.g. hospital occupancy, or case incidence in the country). I would expect reactive closure to be more effective in preventing large waves and more feasible in terms of cumulative policy duration. I would at least recommend including a comment on the possible better outcome of reactive policies. 

6. Synthetic contact matrices specifically derived from highly detailed, country-specific data for many of the countries covered by this study exist (see for instance Mistry et al. 2021). As the matrices contained in the conmat package, as far as I understand, are not directly generated for the single countries, I am wondering whether results would be different when using contact patterns directly obtained from the socio-demographic structure of the population considered, rather than using generalized contact patterns. 

Minor comments 

7. It would be interesting to expand the discussion on the finding that longer cumulative closure duration did not necessarily lead to greater reduction in COVID-19 mortality: for instance, reactive school closure probably would have performed better, and I believe this should be at least acknowledged, if not investigated with the model. 

8. I think it would be important to offer the reader a wider perspective on the existing literature on school closure effectiveness by including more references to previous research. There are many modeling works especially on pandemic influenza but also on COVID-19 that could make a good starting point: see for instance Cauchemez et al. 2008, Wu et al. 2010, Fumanelli et al. 2012, Di Domenico et al. 2021, Rozhnova et al. 2021 only to mention some of them. 

Reviewer #4: a) The semi-mechanistic compartmental model used to simulate COVID-19 transmission is well-structured, with appropriate compartment types. The stratification by age, vaccination status, and virus strain is a robust approach for capturing the dynamics of the pandemic. However, it would be beneficial to consider whether additional stratifications, such as by geographic region within countries, could further enhance the model's granularity.

b) The compartmental model is stratified by age and vaccination status, but it is important to assess whether the chosen age group stratification is appropriate and whether the assumptions regarding vaccination and pre-existing immunity are reasonable. Additionally, are the differential transmissibility and risks associated with each viral variant (wild-type, Delta, Omicron) sufficiently justified?

c) The authors simulate a counterfactual scenario in which schools remained open. It would be helpful if the authors provide detailed information on the assumptions underlying this counterfactual scenario and clearly explained the differences between historical and counterfactual scenarios.

d) Regarding the sensitivity analyses, it is worth evaluating whether these analyses are comprehensive enough to address the potential variability in the model's assumptions. Are there any additional sensitivity analyses, such as varying the number of days schools were closed or the impact of variations in vaccine efficacy, that could strengthen the findings?

e) The decision to use four sequential compartments for both exposed and active states to achieve an Erlang distribution is technically sound, as it avoids the unrealistic assumptions of an exponential distribution for disease progression. However, the authors should clarify whether the number of compartments used is adequately justified. Could fewer or more compartments have yielded similar or better results?

f) The authors employed a Bayesian approach for model calibration, which is a solid choice for managing the complex uncertainties in the model parameters. However, the manuscript should provide more detail on the convergence diagnostics of their chains and include information on how the robustness of their parameter estimates was ensured.

g) The authors selected uniform prior distributions for the primary parameters during calibration. It would be beneficial for the manuscript to justify this choice of priors. Were other distribution types considered, and if so, why were uniform priors ultimately chosen?

h) The authors mentioned the use of random processes to account for unexplained variations in transmission risk. It is important to clarify whether the implementation of random walks and Gaussian updates is appropriate and to provide sufficient reasoning for the frequency of these updates.

i) The model captures changes in social interactions over time using dynamic age-specific mixing matrices, with particular attention to school closures. The manuscript should expand on whether the assumptions around the modification of contact rates, such as the effect of partial school closures on contact rates, are well-supported by empirical data.

j) The use of a convolution process to estimate hospitalisations and deaths from the model's outputs is a practical approach. However, the manuscript should consider whether the convolution process has been adequately validated against empirical data from the analysed countries. Are the assumptions regarding the time distributions for hospitalisations and deaths appropriate and consistent with the literature?

k) The adjustment of infection fatality rates using country-specific multipliers is a necessary step given the variability in health system responses. The method for calibrating these multipliers should be clearly explained and justified in the manuscript.

l) I suggest that the authors include additional validations of their model outputs against independent datasets, such as sub-national data or external validation using datasets not employed in the model calibration.

m) I propose that the authors include more real-world scenarios in their sensitivity analyses, such as varying the effectiveness of non-pharmaceutical interventions or testing the impact of different vaccination rollout speeds.

n) The manuscript should clearly articulate the limitations of the model. For instance, the authors could discuss the potential impact of unmodeled factors such as behavioural changes not captured by mobility data or the effect of partial immunity in populations with mixed infection and vaccination histories.

---

* Please upload any figures associated with your paper as individual TIF or EPS files with 300dpi resolution at resubmission; please read our figure guidelines for more information on our requirements: http://journals.plos.org/plosmedicine/s/figures. While revising your submission, please upload your figure files to the PACE digital diagnostic tool, https://pacev2.apexcovantage.com/. PACE helps ensure that figures meet PLOS requirements. To use PACE, you must first register as a user. Then, login and navigate to the UPLOAD tab, where you will find detailed instructions on how to use the tool. If you encounter any issues or have any questions when using PACE, please email us at PLOSMedicine@plos.org.

* All authors must declare their relevant competing interests per the PLOS policy, which can be seen here:

https://journals.plos.org/plosmedicine/s/competing-interests

For authors with ties to industry, please indicate whether any of the interests has a financial stake in the results of the current study.

FIGURES AND TABLES

SUPPLEMENTARY MATERIAL

REFERENCES

STUDY TYPE-SPECIFIC REQUESTS - MODELLING STUDIES

The following list is derived from Geoffrey P Garnett, Simon Cousens, Timothy B Hallett, Richard Steketee, Neff Walker. Mathematical models in the evaluation of health programmes. (2011) Lancet DOI:10.1016/S0140-6736(10)61505-X: 

* If pertinent, please provide a diagram that shows the model structure, including how the natural history of the disease is represented, the process and determinants of disease acquisition, and how the putative intervention could affect the system.

* Please provide a complete list of model parameters, including clear and precise descriptions of the meaning of each parameter, together with the values or ranges for each, with justification or the primary source cited and important caveats about the use of these values noted.

* Please provide a clear statement about how the model was fitted to the data, including goodness-of-fit measure, the numerical algorithm used, which parameter varied, constraints imposed on parameter values, and starting conditions.

* For uncertainty analyses, please state the sources of uncertainties quantified and not quantified [can include parameter, data, and model structure].

* Please provide sensitivity analyses to identify which parameter values are most important in the model. Uncertainty estimates seek to derive a range of credible results on the basis of an exploration of the range of reasonable parameter values. The choice of method should be presented and justified.

* Please discuss the scientific rationale for the choice of model structure and identify points where this choice could influence conclusions drawn. Please also describe the strength of the scientific basis underlying the key model assumptions.

* For studies that develop a prediction model or evaluate its performance, please ensure that the study is reported according to the TRIPOD statement (https://www.equator-network.org/reporting-guidelines/tripod-statement) and include the completed checklist as Supporting Information. Please add the following statement, or similar, to the Methods: "This study is reported as per the Transparent Reporting of a Multivariable Prediction Model for Individual Prognosis Or Diagnosis (TRIPOD) statement (S1 Checklist)." For studies using machine learning, please use the TRIPOD-AI checklist. When completing the checklist, please use section and paragraph numbers, rather than page numbers.

---

## [Decision Letter · Decision Letter 2]

23 Oct 2024

Dear Dr Ragonnet,

Many thanks for submitting your manuscript "Estimating the impact of school closures on the COVID-19 dynamics in 74 countries: a modelling analysis" (PMEDICINE-D-24-02032R2) to PLOS Medicine. The paper has been seen again by two of the subject experts and a statistician; their comments are included below and can also be accessed here: [LINK]

Thank you for your detailed response to the editors' and reviewers' comments. As you will see, the reviewers are mostly satisfied with your responses to their comments. There are several remaining comments that need to be addressed, and in particular the statistical reviewer has made several points that require substantial detail to be added. After discussing the paper with the editorial team and an academic editor with relevant expertise, we ask you to carefully address the comments in a further revision. We plan to send the revised paper to some or all of the original reviewers.

When you upload your revision, please include a point-by-point response that addresses all of the reviewer and editorial points, indicating the changes made in the manuscript and either an excerpt of the revised text or the location (eg: page and line number) where each change can be found. Please also be sure to check the general editorial comments at the end of the previous decision letter and include these in your point-by-point response. When you resubmit your paper, please include a clean version of the paper as the main article file and a version with changes tracked as a marked-up manuscript. It may also be helpful to check the guidelines for revised papers at http://journals.plos.org/plosmedicine/s/revising-your-manuscript for any that apply to your paper.

We ask that you submit your revision by Nov 13 2024. However, if this deadline is not feasible, please contact me by email, and we can discuss a suitable alternative.

Don't hesitate to contact me directly with any questions (atosun@plos.org). 

Best regards, 

Alexandra 

Alexandra Tosun, PhD 

Associate Editor

PLOS Medicine

atosun@plos.org

Comments from the academic editor:

My overall impression is that the authors have done an excellent job of responding to the reviewers' comments, and that the manuscript is now strong, and appropriately nuanced in its discussion of results.

Comments from the editorial team:

Please note that we always require a point-by-point response to not only reviewer comments, but all editorial comments, including general editorial requests. Please be sure to provide such a document that addresses all editorial comments of the previous decision letter.

Comments from the reviewers: 

Reviewer #2: The authors have satisfyingly addressed my comments. I would like to congratulate the authors for this interesting study. 2 final minor comments:

1) Abstract : « We estimate that school closures achieved moderate to significant burden reductions in most settings over the period 2020-2022. They reduced peak hospital occupancy in nearly all countries, with 72 out of 74 countries showing a positive median estimated effect. However, we identified countries, including multiple European nations and Indonesia, where school closures may have increased overall COVID 19 mortality. » Authors give a proportion of countries (72/74) in which impact is positive on peak hospital occupancy. It would be good to provide a similar statistics in the second sentence discussing the risk of increasing overall mortality.

2) Methods: "Following calibration under the historical scenario, the model simulated a "counterfactual scenario" in which schools would have remained fully open throughout the pandemic." Please clarify what is considered for school holidays? In the counterfactual scenario, was it assumed that schools remain open during school holidays? If the counterfactual scenario is really "schools always open even during holiday periods", the difference in terms of what should be expected for a pattern of standard school year (i.e. when schools are closed during standard holiday terms) should at least be discussed.

Reviewer #3: I thank the authors for their extensive reply to the questions raised by all the reviewers.

I acknowledge the great effort made by the authors in providing more supplementary analyses and comments to justify their modeling choices and improve clarity. I also appreciate that the negative impact of school closure for few countries is less emphasized in the current text than in its previous version.

Overall, I believe that the quality of the manuscript has improved.

- I appreciate the post-hoc statistical analysis on the potential drivers of heterogeneity: it highlights that "young" countries can benefit more from generalized school closures.

- The post-hoc analysis also underlines that, although longer closures are not always associated with a larger number of averted infections and deaths, short closures are rarely highly beneficial. Looking at this and at the results for all the single countries in S2 Appendix, it seems that the timing, duration and type of single school closure events with respect to epidemic peaks may have an impact on the cumulative proportion of cases as well as on deaths. In fact, full closure of schools was not implemented during the Delta phase of the pandemic in many countries where a minor overall impact can be observed.

Another counterfactual scenario with more stringent features for school closure would be very interesting to understand this aspect; however, I recognize that there are already many sensitivity analyses. Therefore, I suggest either implementing this counterfactual scenario at least for a representative country (e.g. Indonesia), and commenting on the possible importance of timing and duration on the outcome of school closure interventions.

- I would also add a comment on accurate testing and contact tracing procedures that may be employed to perform more effective reactive closures at school (or even class) level, in addition to what the authors have already added in the Discussion.

- I would suggest, as future work, to perform further analysis restricted to a small subset of countries (even a single one) for which the pattern of school closures at regional/local level is known, as this would provide more precise insight into the feasibility and expected outcome of this measure.

- Closing schools has generally the effect to reduce to some extent also contacts in other locations, e.g. public transportation: can this be observed in the Google Mobility data?

- I found some misprints.

Line 300: Figures S19-21 -> Figures S25-28.

Caption of Figure S27: India -> Indonesia

Supp A.1.2.5.2: "between 10 and 30% of students attended on-site learning" -> "between 10 and 50%...."

Reviewer #4: a) The authors mention the use of a Bayesian approach with an Adaptive Differential Evolution Metropolis algorithm to calibrate the model. However, to enhance understanding of the calibration process, it would be beneficial to provide more comprehensive details. This should include the priors chosen for all key parameters, the criteria used to determine convergence, and the length of the burn-in period. Additionally, summarising the posterior distributions obtained and discussing the sensitivity of the results to different prior choices would offer readers a clearer picture of how model parameters were estimated and validated.

b) The random perturbation, denoted as W(t), plays a critical role in capturing unobserved variability in transmission. A more thorough explanation of how this random process interacts with other model variables and how it represents interventions not captured by explicit covariates—such as changes in behaviour—would greatly enhance transparency. Furthermore, providing the mathematical form of W(t) and detailing any assumptions regarding its distribution and correlation structure would add clarity and allow for better understanding of how this stochastic element affects the model.

c) Since the model relies on multiple data sources, such as death counts, seroprevalence, and mobility data, it is essential to elaborate on how any missing data or discrepancies between sources were managed. If imputation techniques were employed or if weighting was based on data reliability, these methods should be clearly outlined. A thorough explanation of the data-handling process would provide readers with insight into the model's robustness and reliability.

d) The paper reports various sensitivity analyses, but there is a need for greater detail regarding the framework used. Specifically, it would be helpful to describe the criteria for selecting parameters included in these analyses. Additionally, explaining how the ranges for sensitivity analyses were chosen—whether based on literature, observed data variability, or other factors—would offer context. Addressing whether and how parameter correlations were accounted for during sensitivity analyses is also crucial to provide a complete picture of the approach to assessing model sensitivity.

e) To improve the statistical rigor of the analysis, it is recommended to elaborate on how parameter identifiability was assessed. Were any parameters found to be highly correlated or unidentifiable? Describing such aspects would highlight the challenges encountered during the modelling process. Additionally, outlining how uncertainty in parameter estimates was propagated to model outcomes would help readers assess the robustness and reliability of the model's conclusions. Detailing these approaches ensures transparency and strengthens the overall confidence in the study's findings.

---

## [Decision Letter · Decision Letter 3]

6 Dec 2024

Dear Dr. Ragonnet,

Thank you very much for re-submitting your manuscript "Estimating the impact of school closures on the COVID-19 dynamics in 74 countries: a modelling analysis" (PMEDICINE-D-24-02032R3) for review by PLOS Medicine.

Thank you for your detailed response to the editors' and reviewers' comments. I have discussed the paper with my colleagues and the academic editor, and it has also been seen again by two of the original reviewers. The changes made to the paper were satisfactory to the reviewers. As such, we intend to accept the paper for publication, pending your attention to the editors' comments below in a further revision. When submitting your revised paper, please once again include a detailed point-by-point response to the editorial comments.

[LINK]

In revising the manuscript for further consideration here, please ensure you address the specific points made by each reviewer and the editors. In your rebuttal letter you should indicate your response to the reviewers' and editors' comments and the changes you have made in the manuscript. Please submit a clean version of the paper as the main article file. A version with changes marked must also be uploaded as a marked up manuscript file. Please also check the guidelines for revised papers at http://journals.plos.org/plosmedicine/s/revising-your-manuscript for any that apply to your paper.

We ask that you submit your revision within 1 week (Dec 13 2024). However, if this deadline is not feasible, please contact me by email, and we can discuss a suitable alternative.

Please do not hesitate to contact me directly with any questions (atosun@plos.org). If you reply directly to this message, please be sure to 'Reply All' so your message comes directly to my inbox.

We look forward to receiving the revised manuscript.   

Sincerely,

Alexandra Tosun, PhD

Associate Editor 

PLOS Medicine

plosmedicine.org

Comments from Reviewers:

Reviewer #4: Thank you for thoroughly addressing all my comments and providing detailed clarifications in your revised manuscript. I appreciate the effort you have put into enhancing the rigor and transparency of your analyses.

The manuscript now reflects a robust and valuable contribution to the understanding of the impact of school closures on COVID-19 dynamics. I am happy to recommend this work for publication and look forward to seeing its positive impact in the field.

[LINK]

Requests from Editors:

ABSTRACT

1) l.26: Please change ‘estimate’ to ‘estimated’.

2) l.29, “showing a positive median estimated effect” – Would it be possible to present numerical results here, e.g. as a range of the lowest and highest positive median effect? The same would apply to line 31 and the presentation of COVID-19 mortality.

3) In the last sentence of the Abstract Methods and Findings section, please describe the main limitation(s) of the study's methodology.

4) General guidance for Abstract Methods and Findings (not all may apply):

* Please ensure that all numbers presented in the abstract are present and identical to numbers presented in the main manuscript text.

* Please include the study design, population and setting, number of participants, years during which the study took place, length of follow up, and main outcome measures.

* Please quantify the main results (with 95% CIs and p values).

* Please include the important dependent variables that are adjusted for in the analyses.

AUTHOR SUMMARY

1) l.44: Please replace ‘years’ with ‘stages’.

2) In the final bullet point of 'What Do These Findings Mean?', please include the main limitations of the study in non-technical language.

INTRODUCTION

Please address past research and explain the need for and potential importance of your study. Indicate whether your study is novel and how you determined that. If there has been a systematic review of the evidence related to your study (or you have conducted one), please refer to and reference that review and indicate whether it supports the need for your study.

METHODS AND RESULTS

1) l.113: Oceania' does not appear to be an official description of a continent, which would be Australia. Please revise.

2) ll.133-134: When presenting age, please provide a unit, such as years. Please revise throughout.

3) ll.238-240: Please remove the funding details from the main text. This information should only be included in the online submission form.

4) ll.253-254: “most analysed countries of the European region” – please specify.

5) l.287, please change to: “In the UK, while we estimated that…". Please check for the correct tense throughout. Methods and results should be in the past tense.

6) l.288: “in the over-50” – years old? Please revise.

7) l.327: Please define ‘PHSMs’ at first use.

8) Figure 1: Please change ‘Turkey’ to ‘Türkiye’ and revise throughout.

9) Figure 2: Please change 'Oceania' to 'Australia'. Please note that each figure should be self-explanatory on its own, i.e. we ask you to re-include the panel defining all country abbreviations in Figure 2.

10) Figure 3/4/5: Please define ‘K’, ‘M’ and ‘Jan’ in the figure description. In panel G, please add a unit for age (at minimum in the figure description). Please note that panels C and G are very close together, so we recommend leaving more space between the left and right hand panels.

DISCUSSION

1) l.333: We have noticed that you alternate between using the terms 'peak hospital occupancy', 'peak hospital occupancy pressure' and 'hospital pressure' - please revise throughout. Here, for clarity, we suggest changing 'hospital pressure' to 'hospital occupancy pressure'.

2) l.397: Please temper claims of primacy of results by stating, "to our knowledge" or something similar.

REFERENCES

1) Please ensure that journal name abbreviations match those found in the National Center for Biotechnology Information (NCBI) databases (http://www.ncbi.nlm.nih.gov/nlmcatalog/journals), and are appropriately formatted and capitalised. For example, in reference [12], ‘The Lancet Infectious Diseases’ should read ‘Lancet Infect Dis’.

2) Where website addresses are cited, please include the complete URL and specify the date of access by providing day, month, and year (e.g. [accessed: 12/06/2024]).

SUPPLEMENTARY MATERIAL

1) Please note that all supplementary files should be references in the main text. Please check and revise.

2) In the published article, supporting information files are accessed only through a hyperlink attached to the captions. For this reason, you must list captions at the end of your manuscript file. You may include a caption within the supporting information file itself, as long as that caption is also provided in the manuscript file. Do not submit a separate caption file.

When SI files are contained with a single file:

Please label the file as ‘S1 Supporting Information’.

Please apply alphabetical labelling to each table and figure contained within the S1 file. For example, ‘Fig A’ to ‘Fig Z’ and ‘Table A’ to ‘Table Z’.

Plain text does not need to be labelled and can just be given a title as necessary. For example, ‘Statistical Analysis Plan’.

Please cite tables/figures as ‘Fig A in S1 Supporting Information’ and/or ‘Table A in S1 Supporting Information’, for example.

Please cite plain text as, ‘Statistical Analysis Plan in S1 Supporting Information’, for example.

When SI files are uploaded as separate files:

Please label tables as ‘S1 Table’ (so on) and figures as ‘S1 Fig’ (and so on).

Any additional documents (protocols/analysis plans etc.) can be labelled as ‘S1 Protocol’, for example. Please cite items as exactly as labelled.

SOCIAL MEDIA

To help us extend the reach of your research, please provide any social media handle(s) that would be appropriate to tag, including your own, your co-authors’, your institution, funder, or lab. Please enter in the submission form any handles you wish to be included when we post about this paper.

General Editorial Requests

---

## [Editor Report · Decision Letter 4]

18 Dec 2024

Dear Dr Ragonnet, 

On behalf of my colleagues and the Academic Editor, Peter MacPherson, I am pleased to inform you that we have agreed to publish your manuscript "Estimating the impact of school closures on the COVID-19 dynamics in 74 countries: a modelling analysis" (PMEDICINE-D-24-02032R4) in PLOS Medicine.

I appreciate your thorough responses to the reviewers' and editors' comments throughout the editorial process. We look forward to publishing your manuscript. If you have any questions or concerns during the final steps, please do not hesitate to contact me at atosun@plos.org.

PRESS

Sincerely, 

Alexandra Tosun, PhD 

Associate Editor 

PLOS Medicine